# LEARNING TO EXPLORE WITH IN-CONTEXT POLICY FOR FAST PEER ADAPTATION

## ABSTRACT

Adapting to different peers in multi-agent settings requires agents to quickly learn about the peer's policy from a few interactions and act accordingly. In this paper, we present a novel end-to-end method that learns an in-context policy that actively explores the peer's policy, recognizes its pattern, and adapts to it. The agent is trained on a diverse set of peer policies to learn how to balance exploration and exploitation based on the observed context, which is the history of interactions with the peer. The agent proposes exploratory actions when the context is uncertain, which can elicit informative feedback from the peer and help infer its preferences. To encourage such exploration behavior, we introduce an intrinsic reward based on the accuracy of the peer identification. The agent exploits the context when it is confident, which can optimize its performance with the peer. We evaluate our method on two tasks that involve competitive (Kuhn Poker) or cooperative (Overcooked) interactions with peer agents. We demonstrate that our method induces active exploration behavior, achieving faster adaptation and better outcomes than existing methods.[1]

## 1 INTRODUCTION

Agents in the real world often encounter diverse *peers* in both cooperative and competitive settings. To optimize their outcomes, agents need to observe and adapt to the behaviors of others, as different policies usually have different best response strategies. This **peer adaptation** enables agents to cooperate or exploit effectively in various domains. For example, in board and card games (Brown & Sandholm, 2019; Silver et al., 2017; Hu et al., 2020), players need to adapt to the skill level and style of their opponents or teammates, such as bluffing, cooperating, or signaling. In online multi-player games (Berner et al., 2019), players need to adapt to the changing strategies and tactics of the enemy team, such as counter-picking, ganking, or pushing. In autonomous driving (Jin et al., 2022), agents need to adapt to the changes in traffic conditions and the behaviors of other drivers, such as aggressive or conservative. In service robots, agents need to adapt to the preferences and needs of the users, such as cleaning, cooking, or entertaining. Figure. 1 illustrates an example of this adaptation process. In each episode, conditioned on the experiences (context), the mother eliminates unlikely options and proposes a most probable item to explore the preferences of the baby.

Peer adaptation necessitates the agent's ability to determine the optimal response for each peer. One prevalent approach is opponent modeling (He et al., 2016; Raileanu et al., 2018; Wang et al., 2022; Papoudakis et al., 2021; Yu et al., 2022; Albrecht & Stone, 2018). In this method, the agent learns to model other agents and uses Bayesian inference on state-action pairs to discern the current peer's identity or intention. A significant limitation of this method is its dependence on interaction trajectories. Inadequate exploration with the peer can prevent the unveiling of the peer policy's distinct features. As these methods overlook efficient exploration, they may not perform well in tasks that do not provide direct exploration rewards. Consider a scenario in No Limit Texas Hold'em: when encountering a new opponent, a player should actively probe the opponent's policy, creating diverse game situations and pushing for more showdowns so as to reveal more information. This might involve calling with hands that have a low win probability in some small pots. Although these actions might yield a relatively negative feedback in the short term, the information they deliver is

---

[1]Project page: `https://sites.google.com/view/peer-adaptation`

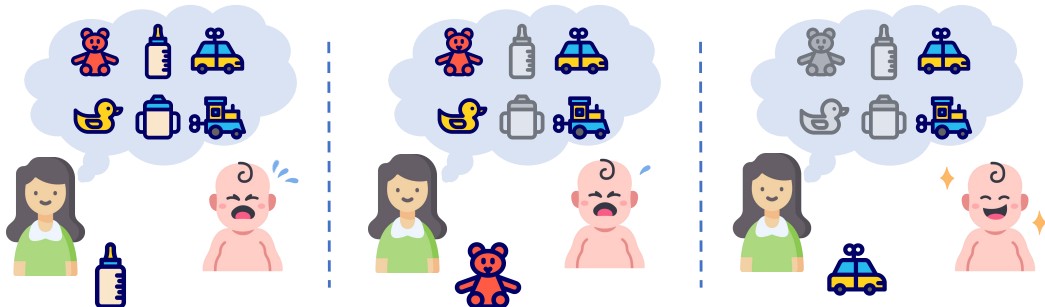

Figure 1: An example of fast peer adaptation with experiences from online interaction, where a mother employs her prior experiences with her baby as contextual cues to determine the appropriate item to offer the baby. In the initial encounter, having observed the baby's disinterest in the milk bottle, the mother infers that the baby is not hungry and suggests a toy as an alternative. Despite the initial unfavorable response to the teddy bear, there is a discernible improvement in the baby's reaction, ultimately leading the mother to successfully choose a toy car in their third interaction.

beneficial for long-term exploitation. However, current algorithms struggle to learn from long-term rewards, as it is challenging to reinforce far earlier exploration actions from the later rewards.

Motivated by the need for conditional adaptation and efficient exploration, we propose *fast Peer Adaptation with In-Context policy* (**PAIC**) that adjusts its behaviour based on its past interactions with a specific peer. Given the context, the policy balances exploration and exploitation to achieve an overall optimal performance during the whole adaptation process. Crucially, fast adaptation entails taking exploratory moves, which do not give instant task rewards but obtain useful information for future exploitation or collaboration. Such exploration behaviors are hard to emerge under regular reinforcement learning (RL) training, as they are easily overwhelmed by behaviors that immediately increase the task reward. Furthermore, the context may span hundreds or thousands of time steps, making it hard to perform credit assignments to exploration moves. To deal with these problems, we introduce peer identification as an auxiliary task that approximates the Bayesian posterior of the peer policy given the observed context. The accuracy of the peer identification serves as an intrinsic exploration reward that boosts exploration.

We conduct experiments in a competitive environment (Kuhn Poker) and a cooperative environment (Overcooked). We show that PAIC adapts faster and achieves higher returns than existing methods when facing novel opponents or collaborators. In further ablation studies, we analyze the effects of the proposed auxiliary task and the corresponding intrinsic reward. A t-SNE visualization of the latent embeddings shows that the PAIC agent quickly recognizes the peer policy in only a few interactions. We also demonstrate the robustness of our method to the different sizes and distribution of the peer training pools.

To summarize, our main contributions are three-fold. 1) We investigate the peer adaption problem and propose an in-context policy that balances exploration and exploitation. 2) We introduce a practical in-context policy learning method that leverages an auxiliary task to efficiently train the policy, while generating intrinsic rewards to encourage exploration. 3) We empirically validate our method in both the competitive card game (Kuhn Poker) and the cooperative game (Overcooked), showing that our in-context policy can quickly adapt to novel peers and achieve high performance.

## 2 RELATED WORK

**ZSC & AHT** Zero-Shot Coordination (ZSC) is first introduced in Other-Play (Hu et al., 2020), referring to the problem of independently training multiple agents of a same algorithm that can cooperate efficiently during test time without additional coordination. Typical self-play algorithms tend to generate agents with arbitrary conventions, making zero-shot coordination impossible. To prevent such conventions, several methods including arbitrary symmetry breaking (Hu et al., 2020), policy hierarchy (Hu et al., 2021; Cui et al., 2021), diverse pool generation (Lupu et al., 2021; Strouse et al., 2021; Zhao et al., 2021) and intrinsic reward (Lucas & Allen, 2022) are proposed. To sum up, ZSC is about generating compatible agents by avoiding arbitrary conventions. But peer adaptation is

about learning to adapt to uncontrollable peers, which may have certain conventions. Therefore, ZSC and peer adaptation are different problems.

A more similar topic is Ad Hoc Teamwork (AHT) (Stone et al., 2010). AHT not only requires no prior coordination, but also assumes no control over teammates (Mirsky et al., 2022), which is different from ZSC and closer to the definition of peer adaptation. Type-based methods (Chen et al., 2020; Barrett & Stone, 2015; Ravula, 2019; Mirsky et al., 2020) predefine a set of types of agents and assume that any encountered new teammate will conform to one of these predefined types. To avoid the drawback of fixed teammate types, recent works leverage graph neural networks (Rahman et al., 2021) and value decomposition (Gu et al., 2021) to generate latent types. Nevertheless, AHT is limited in the domain of teammate collaboration (Mirsky et al., 2022), as it assumes that all agents share a common objective, while still allowing for non-conflicting individual objectives. Instead, peer adaptation does not limit to the cooperation setting, and can be used in both cooperative and competitive scenarios.

**Fast Adaptation.** Fast Adaptation includes adaptation to new environments(tasks) (Raileanu et al., 2020; Zuo et al., 2019) and new agents (Zhu et al., 2021; Zhong et al., 2019; 2021). In this paper, we consider the problem of fast adaptation to peers. Meta-learning methods (Al-Shedivat et al., 2018; Kim et al., 2021) compute meta-multiagent policy gradient during interaction and adapt the policy accordingly. Bayesian inference (Zintgraf et al., 2021) is another method to update the belief about other agents so as to effectively respond to them. Modeling of other agents can help improve the adaptation to them (He et al., 2016; Raileanu et al., 2018; Wang et al., 2022; Papoudakis et al., 2021; Yu et al., 2022; Albrecht & Stone, 2018; Fu et al., 2022; Xie et al., 2021). While our method also includes an auxiliary task for opponent recognition, it does not require opponent observation and executes in a fully decentralized manner. Moreover, none of the previous works consider the challenge of efficient exploration. Our method instead leverages the auxiliary task to generate an intrinsic reward to boost the exploration strategy learning.

**In-Context Policy** In-context learning is initially proposed in the domain of language models (Brown et al., 2020), referring to the ability to infer tasks from context. Recent progress in large language models like GPT (OpenAI, 2023) and LLaMA (Touvron et al., 2023) have shown the potential of in-context learning, where a language model adaptively changes its behavior based on previous inputs (context). Some further extend this term to in-context policy. Algorithm distillation (Laskin et al., 2022) trains a Transformer that captures the pattern of policy improvement during RL training given recent episodes. Several works (Rakelly et al., 2019; Luo et al., 2022) employs contextual policies in meta reinforcement learning to account for changes in environments and tasks. Our method concerns the peer adaptation problem in MARL, instead of the language or single-agent RL domains. Furthermore, a focus of our work is the active exploration during adaptation, which is left unaddressed in previous works.

## 3 METHOD

### 3.1 PROBLEM FORMULATION

In this work, we consider the problem of online adaptation to unknown peers in partially observable environments. Denote the training peer prior distribution as $\Psi$, from which peer agent policies $\psi$ can be sampled. We assume unknown but stationary peer agents, which can be treated as parts of the environment dynamics for the ego agents. In this way, the environment from the perspective of the ego agents can be seen as a decentralized partially observable Markov decision process (Dec-POMDP) (Bernstein et al., 2002) $(\mathcal{S}, \{\mathcal{O}_i\}, \{\mathcal{A}_i\}, \mathcal{P}, R, \gamma)$, where $\mathcal{S}$ is the state space, $\mathcal{O}_i, \mathcal{A}_i$ are the observation and action spaces for ego agent $i$, $\mathcal{P}$ is the state transition probability, $R$ is the shared reward function for all ego agents, and $\gamma$ is the discount factor. The state space $\mathcal{S}$ contains the peer agents $\psi$, sampled from the peer pool $\Psi$ during the sampling of the initial state and kept unchanged throughout the adaptation procedure. However, the identities of the peer agents are usually not directly revealed to the ego agents in the observation spaces $\{\mathcal{O}_i\}$, forcing the ego agents to deduce this information from their observations. For simplicity, we hereafter consider a version with one ego agent (agent 1) and one peer agent. Adding more peer agents does not alter the problem setting as peer agents are subsumed into the environment dynamics, while more ego agents can be handled by applying standard multi-agent reinforcement learning techniques, e.g. centralized training and decentralized execution (CTDE).

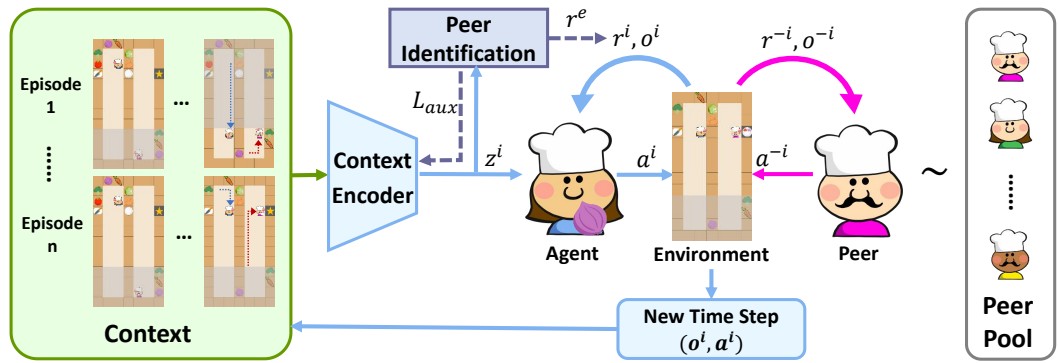

Figure 2: Illustration of the in-context policy for peer adaption. The policy of the PAIC agent (left) is conditioned on the current context, which consists of the self-observation sequences from the previous episodes, and generates the trajectory for the next episode. During training, the PAIC agent interacts with peer agents (right) from a training peer pool, ensuring that it can adapt to diverse behaviors. Peer identification task guides the encoder and generates reward to encourage exploration.

In real-world scenarios, there are often repeated interactions between a group of agents. The agents seek to optimize not the return of any single episode, but the cumulative return over a number of episodes. As such, we define the goal of **peer adaptation** over $N_{\text{ctx}}$ episodes as the following:

$$\text{maximize} \quad \mathbb{E}[\sum_{n=1}^{N_{\text{ctx}}} \sum_{t=1}^{T_n} \gamma^{t'} r_{n,t}^1] \tag{1}$$

where $r_{n,t}^1$ is the reward for the ego agent at time step $t$ of episode $n$, $T_n$ is the length of episode $n$, $t' := t + \sum_{n'=1}^{n-1} T_{n'}$ is the cumulative number of time steps until time step $t$ of episode $n$. This peer adaptation objective is amenable to standard RL algorithms by concatenating $N_{\text{ctx}}$ episodes together and computing the joint returns, similar to the multi-episode return objective used in (Xie et al., 2021).

### 3.2 IN-CONTEXT POLICY

Multi-agent reinforcement learning (MARL) often involves online interactions with other agents, whose identities may be unknown or uncertain. To cope with this challenge, an agent needs to leverage its observed trajectories of states and actions, referred to as **context**, to infer the type and mind (e.g., belief, intention, desire) of its peer agent. Existing works on opponent modeling either ignore the context across episodes (He et al., 2016; Papoudakis et al., 2021), or decouple the modeling from the policy learning (Fu et al., 2022). However, these approaches may suffer from distributional shifts if the ego policy used for online adaptation differs from the one used for offline modeling. This could result in inconsistent or suboptimal behaviors from both agents, especially in hard scenarios where active exploration is needed to model the peer. Therefore, we argue that the opponent modeling module should be jointly trained with the policy in an online fashion, to align the behaviors and expectations of both modules.

To overcome these problems, we propose learning an end-to-end **in-context policy** $\pi(a|o, C)$ that outputs an action distribution given the current observation and a context $C = \{\{o_{n,t}^1, a_{n,t}^1\}_{t=1}^{T_n}\}_{n=1}^{N}$ of $N$ trajectories. The context also includes the current episode $N$, which may be incomplete. In this case, $T_N$ is the current number of steps in episode $N$. The number of episodes in the context $C$ may vary, and so do the lengths of the episodes in a single context. We build a **context encoder** $\chi$ parameterized by $\theta$ to encode contexts with varying sizes to a fixed-length vector $z^1 \in \mathbb{R}^m$:

$$z^1 := \chi_\theta(C) = g_\theta \left( \frac{1}{N} \sum_{n=1}^{N} \frac{1}{T_n} \sum_{t=1}^{T_n} f_\theta(o_{n,t}^1, a_{n,t}^1) \right) \tag{2}$$

where $f_\theta : \mathbb{R}^{|\mathcal{O}^1|} \times \mathbb{R}^{|\mathcal{A}^1|} \to \mathbb{R}^m, g_\theta : \mathbb{R}^m \to \mathbb{R}^m$ are MLPs, $\chi(\emptyset) := \mathbf{0}$. By taking mean over the time steps in the context, we improve the scale invariance of the context encoder.

This architecture choice for context encoder treats the context as a set of state-action pairs, which ignores intra- and inter-episode temporal order. This may limit its performance in environments where the order is crucial for identifying the peer agent. Alternatively, one could use order-aware architectures, such as RNNs or transformers with positional encoding, to encode the context. However, these architectures may face difficulties in learning long-term dependencies and preventing overfitting.

With the context encoded by $\chi$, an in-context policy $\pi_\theta(a|o, \chi_\theta(C))$ is trained to maximize the peer adaptation objective 1. The encoder and the policy are jointly parameterized and optimized in an end-to-end manner using PPO (Schulman et al., 2017). By end-to-end training of both components, the encoder $\chi_\theta$ is trained to model trajectories sampled by the current policy $\pi_\theta$, eliminating the distribution mismatch.

## 3.3 Peer Identification

For adaptation to an unknown peer, the ego agent must first infer certain characteristics of the peer agent, e.g. strategies and preferences, before the best response can be determined and carried out. However, this inference may require a sophisticated exploration strategy, which is hard to learn directly from the original task reward. To overcome this issue, we propose **peer identification** as an auxiliary task to the encoder and use it to generate additional exploration rewards for the ego agent.

Formally, the encoder is trained to approximate the Bayesian posterior over the peer type given the context $C$:

$$p(\psi|C) = \frac{p(C|\psi)p(\psi)}{\sum_{\psi'} p(C|\psi')p(\psi')} \tag{3}$$

Bayesian inference is a common technique for opponent modeling in the literature. Here we use it implicitly as an auxiliary task to learn the representation, rather than explicitly using it in inference as in (Fu et al., 2022; Yu et al., 2022). For a peer pool $\Psi$ containing a finite number of peer policies, the Bayesian posterior can be approximated by minimizing the following cross-entropy loss:

$$L_{\text{aux}}(\theta) = \mathbb{E}_{(\psi_i, C_i)} \left[ CE(h_\theta(\chi_\theta(C_i)), i) \right] \tag{4}$$

where $h_\theta : \mathbb{R}^m \to [0, 1]^{|\Psi|}$ produces the posterior distribution from encoder outputs, $C_i$ is the context formed by $\pi_\theta$ interacting with $\psi_i$. We optimize $L_{\text{aux}}$ using the same mini-batch used in RL training; see Algorithm 1 for details.

After obtaining the peer identifier $h_\theta$, we use the probabilities it outputs to generate an exploration reward and encourage the ego agent to be more certain about the identity of the peer. We set

$$r^e := h_\theta(\chi_\theta(C_i))_i \tag{5}$$

so the agent is rewarded for assigning more probability to the correct peer, $r^e \in [0, 1]$. The final reward for the ego agent is computed as $r^1 = r + c \cdot r^e$, where $r$ is the original task reward and $c$ is a coefficient. During training, we linearly decays $c$ from $c_{\text{init}}$ to 0 in $M$ environment steps.

## 3.4 Training Procedure and Online Adaptation

See Algorithm 1 for our training algorithm. During training, we maintain a context for every peer agent in the training peer pool $\Psi$. The context receives a new observation-action pair at every time step during policy rollout (Line 10) and gets cleared after reaching $N_{\text{ctx}}$ episodes (Line 12). During policy rollouts, the intrinsic exploration reward is generated and added to the task reward (Line 8). After collecting a batch of data, Line 16 updates the actor and critic for RL and the context encoder simultaneously. For online adaptation, we also maintain a context of no more than $N_{\text{ctx}}$ episodes and update it at every time step. The intrinsic reward is not used in online adaptation.

---

**Algorithm 1** Training Procedure of PAIC

---

**Require:** Training peer pool $\Psi$, context size $N_{\text{ctx}}$
 1: $\forall \psi_i \in \Psi, C_{\psi_i} \leftarrow \emptyset$                 $\triangleright$ Initialize training contexts
 2: Randomly initialize $\theta$
 3: **while** Maximum training step not reached **do**
 4:      $D \leftarrow \emptyset$                 $\triangleright$ Initialize current training batch
 5:      **while** Current batch size not reached **do**
 6:          **for all** $\psi_i \in \Psi$ **do**             $\triangleright$ Policy rollout
 7:              Take an action with $a_t^1 \sim \pi_\theta(a|o_t^1, \chi_\theta(C_{\psi_i}))$ and $\psi_i$, obtain $r_t, o_{t+1}^1$
 8:              Compute $r_t^1$ using Eq. 5 and $r_t^1 = r_t + c \cdot r_t^e$
 9:              Put $(C_{\psi_i}, i, o_t^1, a_t^1, r_t^1)$ into $D$
10:             Update $C_{\psi_i}$ with $(o_t^1, a_t^1)$
11:             **if** $C_{\psi_i}$ has $N_{\text{ctx}}$ completed episodes **then**
12:                $C_{\psi_i} \leftarrow \emptyset$
13:             **end if**
14:          **end for**
15:      **end while**
16:      Update $\theta$ with PPO loss and $L_{\text{aux}}$ using $D$
17: **end while**

---

## 4 EXPERIMENTS

In this section, we conduct experiments in two typical environments, Kuhn Poker (Competitive) and Overcooked (Cooperative), to answer the following questions: 1) How well can PAIC exploit the opponent in the competitive setting? 2) How well can PAIC adapt to the partner in the cooperative setting? 3) How do peer identification and the intrinsic reward influence learning and adaptation? 4) How sensitive is PAIC to the training pool? 5) What does PAIC learn in its latent space?

### 4.1 EXPERIMENT SETUP

To demonstrate the validity of PAIC, we conduct online adaptation experiments in two commonly used environments in the field of MARL: Kuhn Poker (Kuhn, 1950) and Overcooked (Carroll et al., 2019). The environments encompass a wide spectrum of scenarios, characterized by diverse aspects such as cooperative and competitive settings, partial observability, as well as short and long time horizons. We use the average episodic rewards or success rates (see below) over $N_{\text{ctx}}$ episodes as the evaluation metric.

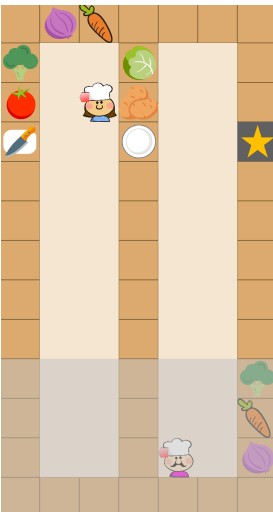

**Kuhn Poker** (Kuhn, 1950) is a simplified two-player poker game involving a deck of three cards and at most three rounds. Both players place an ante of 1 before any action, and each player is dealt one private card initially. The players, P1 and P2, take turns to decide whether to Bet (Call) or Pass (Check). The game ends when one of the players unilaterally passes and forfeit the pot to the other player. If neither or both of the players pass, the game ends in a showdown, where the hands of the players are revealed to each other to decide the winner. In this paper, the ego agent is P1 while the peer agent plays P2. The peer agent pool is generated in a similar manner as in (Southey et al., 2009), which parameterizes the P2 policy space with two parameters.

Figure 3: The partially observable multi-recipe overcooked environment. The activity in the masked area is currently invisible to the left agent.

**Overcooked** (Carroll et al., 2019) is a collaborative cooking game where agents, acting as chefs, work together to complete a series of sub-tasks and serve dishes. To add to the challenge and promote diverse policy behaviors, we employ a partially observable multi-recipe version of Overcooked based on (Charakorn et al., 2023), shown in Fig 3. Our scenario features a total of 6 kinds of ingredients and 9 recipes. The game environment includes a series of counters that divide the room, compelling the agents to collaborate by passing objects, including ingredients and plates, over the counter. The ego agent is the left agent in charge of making dishes while the peer agent delivers them at the right. There is

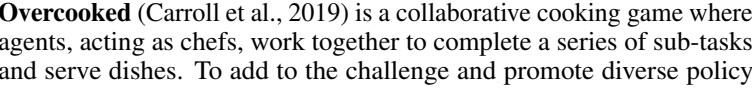

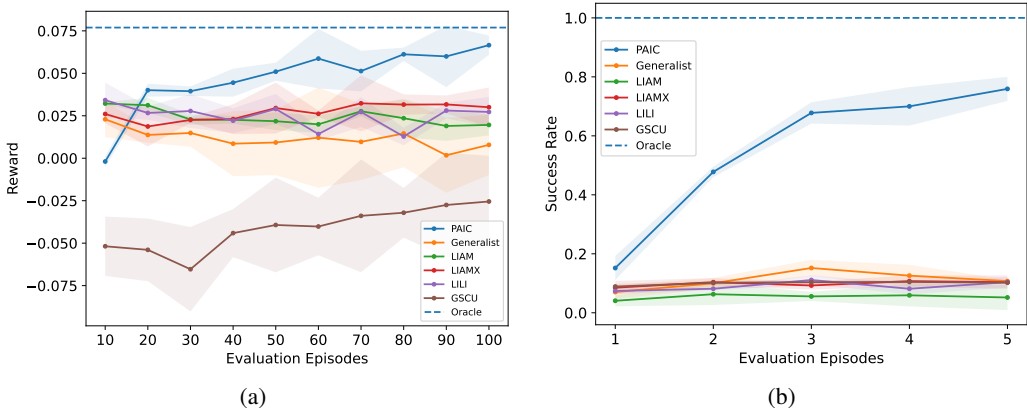

(a)                                                                    (b)

Figure 4: The online adaptation results on Kuhn Poker (a) and Overcooked (b). PAIC continuously improves over the whole online adaptation process, outperforming baselines in both environments. In particular, PAIC is the only agent capable of adaptation in the Overcooked environment. Oracle denotes best responses designed separately for every peer in the test pool.

Table 1: The average reward of PAIC and baselines over 100 online test episodes in Kuhn Poker.

| Methods | PAIC | Generalist | LIAM | LIAMX | LILI | GSCU |
|---------|------|-----------|------|-------|------|------|
| Reward | **0.047 ± 0.004** | 0.012 ± 0.016 | 0.024 ± 0.001 | 0.027 ± 0.008 | 0.025 ± 0.001 | -0.041 ± 0.021 |

also a horizontal wall across the room, blocking the line of sight of the agents, forcing them to move across the wall to see the other side.

Existing approaches (Strouse et al., 2021; Charakorn et al., 2023; Lupu et al., 2021) primarily employ Reinforcement Learning algorithms in conjunction with diversity objectives to train policies that exhibit a wide range of behaviors. In this paper, however, we adopt a different approach by constructing a collection of rule-based peer policies as the right agent, each representing a specific preference for ingredients and recipes. The rule-based agents are designed to take and serve only the ingredients and dishes that match their preferences. We leverage these preference-based policies to capture more human-like behavior within the game.

We choose the following baselines to validate the adaptation and peer modeling capability of PAIC.

- GSCU (Fu et al., 2022) trains a conditional policy conditioned on a pre-trained opponent model. It is notable that GSCU assumes the availability of peer observations and actions after the end of an episode, which is not necessary for PAIC.
- LIAM (Papoudakis et al., 2021) models the opponent's observation and action as an auxiliary task under partially observable settings.
- LIAMX is a variant of LIAM with cross-episode contexts.
- LILI (Xie et al., 2021) models the transitions observed by the ego agent using the last episode, implicitly encoding the opponent as environment dynamics for the ego agent.
- Generalist is a recurrent policy with access to cross-episode contexts, trained using vanilla PPO.

### 4.2 HOW WELL CAN PAIC EXPLOIT THE OPPONENT IN THE COMPETITIVE SETTING?

In Kuhn Poker, we randomly sample 40 *P2* policies from the parameterized policy space as the training pool. We also sample another 10 different policies for online adaptation testing. The online adaptation procedure spans 100 episodes.

The average rewards every 10 episodes during the online adaptation are presented in Figure 4(a), with average rewards over all 100 episodes shown in Table 1. Standard deviations are reported over

Table 2: Average success rates of PAIC and baselines over 5 online test episodes in Overcooked.

| Methods | PAIC | Generalist | LIAM | LIAMX | LILI | GSCU |
|---|---|---|---|---|---|---|
| Success Rate | **0.553 ± 0.029** | 0.111 ± 0.016 | 0.054 ± 0.027 | 0.099 ± 0.009 | 0.090 ± 0.007 | 0.100 ± 0.005 |

3 training seeds. PAIC can be seen to outperform the baselines and continuously improve over the course of adaptation, demonstrating effective and efficient exploitation behavior. In particular, we note that the PAIC agent also explicitly opts to explore the peer's strategy by using a more aggressive strategy for the first 10 episodes. During this time, the game enters the showdown phase at a higher rate of $\sim 0.64$ for the PAIC agent, so it gets to see the peer's hand more frequently and obtains more information about the peer's strategy. In comparison, the showdown rate for LILI holds steady at $\sim 0.60$ over all of the 100 episodes of interaction. GSCU also improves along the online interactions but fails to reach a satisfying level of rewards within the testing time horizon, due to a low starting point. During online adaptation, GSCU may at times use a "conservative policy" that is the Nash Equilibrium policy in Kuhn Poker. This NE policy does not actively try to exploit its opponent, leading to a relatively unsatisfactory performance at first. Other baselines mainly fluctuate around the initial performance, showing that it is hard to make use of the context without proper guidance.

### 4.3 How well can PAIC adapt to the partner in the cooperative setting?

The Overcooked environment poses a significant challenge for adaptation. The peer agent is located in the right side of the room, inaccessible to the ego agent, forcing them to work together. The context consists of a small number of episodes ($N_{\text{ctx}} = 5$), each lasting for dozens of steps. However, as the peer agent only touches ingredients and dishes within its preference and ignores everything else, most of the context contains little information about its true preference. This requires the ego agent to actively perform exploratory actions. We sample 18 policies from the training pool and another 9 policies from the testing pool.

Figure 4(b) and Table 2 show that PAIC is the only agent that can adapt to the peer in the Overcooked environment and all of the baselines fail. Standard deviations are reported over 3 training seeds. Specifically, while GSCU can adapt to its peers and keep improving in Kuhn Poker, it fails in Overcooked due to the lack of an effective exploration strategy. The conservative policy of GSCU in Overcooked is a generalist policy that rarely serves the preferred dish. Consequently, the peer stands still for most of the time, revealing little information for GSCU to model. The results also demonstrate that the context encoder $\chi$ can efficiently summarize long-term contexts and capture only the useful portion.

### 4.4 How do the auxiliary task and reward influence learning and adaptation?

Here we examine the effect of the auxiliary task and reward on the training process and final convergence. Figure 5 contains the training success rates for PAIC and ablations without the auxiliary reward or task. PAIC-reward-aux means that we ablate the auxiliary peer identification task as well as the exploration reward. PAIC-reward means that we only ablate the exploration reward but reserve the auxiliary task. It can be observed that the performance drops severely after removing the auxiliary reward, indicating that the reward is critical for performance. Without the exploration reward, the agent almost never goes down to the lower room. In comparison, the full PAIC agent visits the lower room once on average in an online adaptation procedure. Further removing the auxiliary task also hurts performance and causes training instabilities.

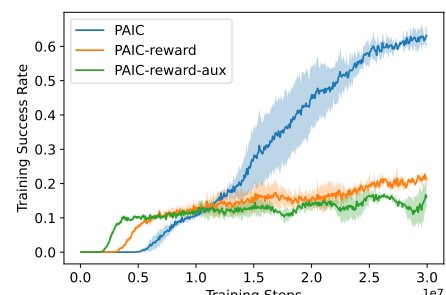

Figure 5: Training success rates for PAIC ablations on Overcooked.

### 4.5 How sensitive is PAIC to the training pool?

To further assess the robustness of our method, we conduct evaluations using varied training and testing pools within the Overcooked task. We define three sizes for the training pools. For each size, we generate three unique training pools and three corresponding testing pools. Each training pool comprised $K$ policies, whereas each testing pool contained 9 policies. Table 3 presents the adaptation performance of PAIC agents corresponding to various training pool sizes. For each pool size, the

Table 3: Success rates of PAIC on different sizes of training pool in Overcooked

| Training Pool Size | K=9 | K=18 | K=36 |
|---|---|---|---|
| Success Rate | $0.510 \pm 0.036$ | $0.573 \pm 0.051$ | $0.587 \pm 0.048$ |

success rate presented is the average derived from evaluations across three distinct sets of training and testing pools. The low variance of success rate shows that PAIC is robust to different pools. Notably, there is a slight increase in the success rate with the enlargement of the pool size, attributable to the augmented volume of peer policy data. Despite minor variances in success rates, PAIC agents trained across all pool sizes consistently and significantly surpassed the baseline performance delineated in Table 2, confirming the robustness of our proposed method.

### 4.6 WHAT DOES PAIC LEARN IN ITS LATENT SPACE?

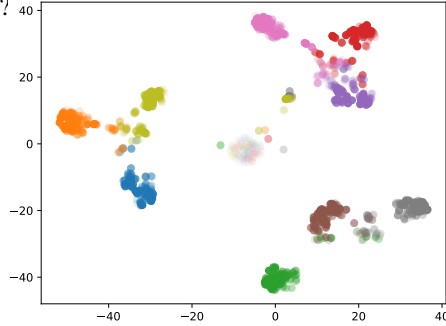

We visualize the latent embeddings in Overcooked produced by the context encoder $\chi$ over the course of online adaptation. Fig. 6 shows the projection of latent embeddings $z$ to 2 dimensions using t-SNE, where each color corresponds to a testing peer. The transparency of the dots indicates the time order in online adaptation. Initially, when the context is uninformative, latent embeddings for all peer policies are similar to each other, as seen in the center of the plot. As the adaptation progresses, the latents first split into three broad clusters, then nine clusters of different colors, showing that the context encoder recognizes the peers.

Figure 6: t-SNE plot of the latent embeddings produced by the PAIC encoder in Overcooked.

## 5 CONCLUSION

In this paper, we propose fast peer adaptation by learning in-context policy (PAIC), a method for training agents that adapt to unknown peer agents efficiently. Autonomous agents in the real world observe and adapt to the behaviors of their peers, whether to facilitate cooperation or exploitation, namely **peer adaptation**. To achieve this goal, agents need to strike the right balance between exploration and exploitation, identifying the peer policy before taking the appropriate response. PAIC learns an in-context policy that adaptively changes its behavior as the interaction progresses. Conditioned on its own observation and action trajectories, PAIC policy explores the peer agent when the context is uncertain, and exploits or collaborates otherwise. Peer identification is proposed as an auxiliary task, generating an exploration reward that encourages the PAIC agent to know better about its peers, thus improving the overall performance. We conduct experiments in Kuhn Poker and Overcooked, two popular MARL environments covering a wide range of properties. Experimental results confirm that the PAIC agent can efficiently adapt its policy based on the context, achieving good performance in both cooperative and competitive environments.

**Limitation:** There are certain limitations with PAIC. We only consider purely cooperative and purely competitive environments. It would be interesting to see whether PAIC can handle more complex mixed-motive environments, where the agents have to balance their own interests and the collective welfare. Another limitation is that we assume that the peer agent does not update its policy during test time. However, in the real world, peers, such as humans, may be able to tune their policies online. How to adapt to such non-stationary peers is an interesting direction for future work.

**Future Work:** A key future direction is peer pool generation. The performance of PAIC agents relies heavily on the diversity of the behaviors in the peer pool. Generating more complex and sophisticated strategies for the training peer pool could further enhance the performance of PAIC. Alternative auxiliary tasks may also benefit policy learning. Moreover, extending our approach to embodied AI scenarios, where the agents have to perceive and act in complex 3D environments (Qiu et al., 2017), could open up new possibilities and challenges for online peer adaptation. Exploring cases where the peer agent is also adapting can be both more challenging and more promising for applications in the real world. Another important direction is to conduct human studies to evaluate how PAIC agents can interact with human peers in various settings. This would require addressing issues such as human factors, ethical considerations, and user feedback.

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

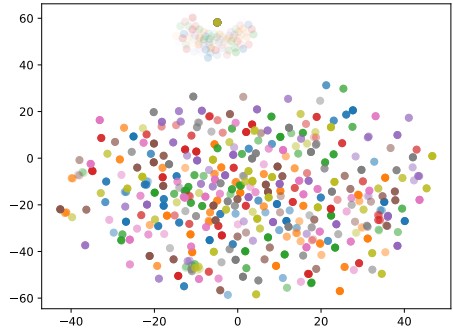

Figure 7: t-SNE projections of latent embeddings produced by LILI.

## A  ADDITIONAL VISUALIZATION

Fig. 7 is a visualization of latent embeddings generated by the encoder of LILI (Xie et al., 2021). Other than the initial cluster with an empty context, the embeddings scatter around the space with no discernible pattern, indicating that the encoder is unaware of the peer's identity.

## B  ENVIRONMENT DETAILS

### B.1  KUHN POKER

Kuhn Poker is a simplified two-player (P1 & P2) poker game (Kuhn, 1950). The game involves a deck of three playing cards, where each player is dealt one card. The cards are ranked (from lowest to highest) Jack, Queen, King. There are no suits in Kuhn poker, only ranks. And the action is restricted to bet and pass, different from No Limit Texas Hold'em, which supports multi-round raising. The game proceeds as follows:

1. Both players put one ante (chip) into the pot.
2. Each player is dealt with one card from the deck. The remaining card is unseen by both players.
3. After the deal, P1 is the first to take action, choosing to bet 1 chip or pass.
   - If P1 chooses to bet, then P2 can bet (call P1's bet and game ends in a showdown) or pass (fold and forfeit the pot).
   - If P1 chooses to pass, then P2 can pass (check and game ends in a showdown) or bet.
   - If P2 bets after P1 passes, P1 should choose to bet (call P2's bet and game ends in a showdown) or pass (fold and forfeit the pot).

In this paper, we focus on learning the adaptation strategy of P1 against P2, and the peer plays as P2. Following the strategy simplification approach introduced by Southey et al. (Southey et al., 2009), we eliminate obviously dominated policies for P2. For example, P2 never bets with Queen after P1 checks, because P1 will always fold with Jack and always call with King. The whole simplified game tree can be found in the paper (Southey et al., 2009). This simplification allows us to parameterize the P2 policy using two parameters $(\xi, \eta)$ within the range of 0 to 1. $\eta$ is the probability of betting Queen after P1 bets. $\xi$ is the probability of betting Jack after P1 passes. Consequently, the entire policy space of P2 can be divided into six sections, each corresponding to a best response from P1.

**Observation Space.** The agents in the game observe a state represented by a 13-dimensional vector, consisting of 3 one-hot vectors. The first one-hot vector is a 7-dimensional representation of the current stage in the game tree. The second and third one-hot vectors, each of 3 dimensions, represent the hand card of the ego player and the opponent. If it has not come to a showdown stage, the opponent hand is always represented by a all-zero vector.

**Action Space.** As stated above, each player can only choose to bet or pass, so the action space is a discrete space with 2 actions.

**Reward.** The reward is not directly determined by the pot itself. Instead, it is calculated based on the chips present in the pot minus the chips contributed by the winner. For the loser, the reward is the negative value of the winner's reward. If the game ends in a showdown, the player holding the highest-rank card wins the pot. If no player bets, then the pot is 2, so the reward is $\pm 1$. Otherwise the pot is 4 (one player bets and the other one bets thereafter), so the reward is $\pm 2$. If the game ends due to one player forfeiting the pot, then the other player wins the pot of 3 chips, so the reward is $\pm 1$.

### B.2 OVERCOOKED

Overcooked is a collaborative cooking game where players take on the roles of chefs working together to complete various sub-tasks and serve dishes (Carroll et al., 2019). In this paper, we introduce a more complex Multi-Recipe version, which builds upon the modifications by Charakorn et al. Charakorn et al. (2023). Specifically, we add two extra ingredients, potato and broccoli, and correspondingly more recipes to increase the challenge and encourage diverse policy behaviors. The game scenario involves a total of 6 ingredients (Tomato, Onion, Carrot, Lettuce, Potato, and Broccoli) and 9 recipes. Notably, the game environment features a counter that divides the room, necessitating collaboration between chefs as they pass objects such as ingredients and plates back and forth over the counter. To serve a dish, the necessary ingredients should be first taken to the cut board and chopped. After all the required ingredients are chopped and put into a plate, the dish need to be carried to the deliver square to finish the task. Furthermore, we add partial observability to the game, separating the game scene horizontally into an upper room and a lower room. Each agent can only see objects in the same room as itself.

**Observation Space.** The observation is a 105-dimensional vector. It consists of multiple features, including position, direction, holding objects, front objects and so on. There is a flag for each relevant object that indicates its visibility to account for partial observation.

**Action Space.** Each agent can choose from a discrete space with 6 actions: move left/right/up/down, interact (with objects) and no-op (take no action).

**Reward.** Overcooked is a fully cooperative game, so all agents share the reward. There are three types of rewards in the game. The first is interact reward. Each agent receive a reward of 0.5 if an object is interacted by an agent. Note that repeated interaction with a same object do not accumulate additional rewards. The second is progress reward. Each agent receives a reward of 1.0 when the state of a recipe progresses. For example, if a chopped carrot is placed into a plate, transitioning the recipe state from "chopped carrot" to "carrot plate," each agent is rewarded. The third is complete reward. When a dish satisfying a recipe is served to the deliver square, each agent receives a reward of 10.0.

## C PEER POOL GENERATION

In PAIC pipeline, we need to first collect a diverse peer pool $\Psi$, which contains representative behaviors of the real peer distribution. Current methods (Strouse et al., 2021; Charakorn et al., 2023; Lupu et al., 2021) mainly use RL algorithms with diversity objectives to train policies that exhibit various behaviors. In this paper, however, we generate a collection of rule-based policies. It is because the P2 policy in Kuhn Poker can be parameterized by two probabilities $(\xi, \eta)$. And we believe the preference-based policies in overcooked capture more human-like behaviors within the game. The details of the rule-based policy pool are listed below.

### C.1 KUHN POKER

As we mentioned above, we eliminate the dominated strategies for P2. Therefore, P2 policy can be determined by two factors: $\eta$ and $\xi$. $\eta$ is the probability of betting with Queen after P1 bets. $\xi$ is the probability of betting with Jack after P1 passes.

In this way, we can easily generate as many P2 policies as we want by randomly sampling $\xi$ and $\eta$. In this paper, we sample 40 P2 policies for training and 10 P2 policies for testing.

Table 4: Hyperparameters for the Kuhn Poker environment.

| Parameter Name | Algorithms | | | | |
|---|---|---|---|---|---|
| | PAIC | Generalist | LILI | LIAM(X) | GSCU |
| Learning Rate | 2e-4 | 2e-4 | 2e-4 | 2e-4 | 5e-4 |
| PPO Clip $\epsilon$ | 0.2 | 0.2 | 0.2 | 0.2 | 0.2 |
| Entropy Coefficient | 5e-4 | 5e-4 | 5e-4 | 5e-4 | 0.01 |
| $\gamma$ | 0.99 | 0.99 | 0.99 | 0.99 | 0.99 |
| GAE $\lambda$ | 0.95 | 0.95 | 0.95 | 0.95 | 0.95 |
| Batch Size | 80000 | 80000 | 80000 | 80000 | 1000 |
| # Update Epochs | 15 | 15 | 15 | 15 | 5 |
| # Mini Batches | 12 | 12 | 12 | 12 | 30 |
| Gradient Clipping (L2) | 2.0 | 2.0 | 2.0 | 2.0 | 0.5 |
| Activation Function | ReLU | ReLU | ReLU | ReLU | ReLU |
| Actor/Critic Hidden Dims | [128, 128] | [128, 128] | [128, 128] | [128, 128] | [128, 128] |
| $f_\theta$ Hidden Dims | [64, 64] | N/A | N/A | N/A | N/A |
| $g_\theta$ Hidden Dims | [64, 64] | N/A | N/A | N/A | N/A |

## C.2 OVERCOOKED

In this paper, we generate preference-based peer agents that possess individual preferences for specific recipes. For instance, each peer agent may have a preference for a recipe such as Tomato & Onion Salad. These peer agents are consistently positioned on the right side of the kitchen and interact exclusively with ingredients and dishes that align with their preferred recipe. For instance, a Tomato & Onion Salad peer agent focuses on sub-tasks related to handling Tomato and Onion ingredients (chopped or fresh) or delivering dishes that exclusively contain these two ingredients.

At each time-step, the agent evaluates whether its current sub-task is completed or not. If the sub-task remains unfinished, the agent determines the shortest path to the target position and navigates accordingly. On the other hand, if the sub-task is completed, the agent samples a new sub-task from its preferred set of sub-tasks.

In addition, there are two parameters that control more fine-grained strategies. $P_{nav}$ is the probability of moving right/left instead of up/down when there are multiple shortest paths. $P_{act}$ is the probability of choosing a random action instead of the optimal action for the current sub-task. For example, suppose the peer is trying to put the Tomato onto the counter. With probability $P_{act}$, it randomly chooses an action from the action space. With probability $1 - P_{act}$, it chooses the optimal action (navigate or interact).

We believe such rule-based agents exhibit behaviors that are more human-like than self-play agents trained by RL algorithms. First, cognition studies (Etel & Slaughter, 2019; Sher et al., 2014) suggest that humans indeed act based on intentions and desires. Furthermore, self-play agents often have arbitrary conventions (Hu et al., 2020). In overcooked, such conventions may be putting/taking ingredients and plates at a certain counter, and refuses to interact with objects at different locations. However, these self-play conventions rarely appear in human behaviors. As a result, preference-based policy is a better choice.

The overcooked scenario in this paper consists of 9 recipes. There are also two parameters $P_{nav}$ and $P_{act}$ that control more fine-grained strategies. When generating a new peer policy, we first uniformly sample its preferred recipe from the 9 recipes, and then randomly sample $P_{nav}$ and $P_{act}$. The training peer pool contains 18 policies and the testing peer pool contains 9 policies.

## D TRAINING DETAILS

The general training pipeline of LILI, LIAM, LIAMX, and Generalist is similar to PAIC in Algorithm 1. The difference between these methods and PAIC is that they have some different auxiliary tasks and do not have the exploration reward used in PAIC. The training procedure of GSCU is quite different from other methods, which can be found in the original paper.

Table 5: Hyperparameters for the Overcooked environment.

| Parameter Name | Algorithms | | | | |
|---|---|---|---|---|---|
| | PAIC | Generalist | LILI | LIAM(X) | GSCU |
| Learning Rate | 1e-3 | 1e-3 | 1e-3 | 1e-3 | 7e-4 |
| PPO Clip $\epsilon$ | 0.2 | 0.2 | 0.2 | 0.2 | 0.2 |
| Entropy Coefficient | 0.03 | 0.03 | 0.03 | 0.03 | 0.01 |
| $\gamma$ | 0.99 | 0.99 | 0.99 | 0.99 | 0.99 |
| GAE $\lambda$ | 0.95 | 0.95 | 0.95 | 0.95 | 0.95 |
| Batch Size | 72000 | 72000 | 72000 | 72000 | 2500 |
| # Update Epochs | 4 | 4 | 4 | 4 | 8 |
| # Mini Batches | 18 | 18 | 18 | 18 | 2 |
| Gradient Clipping (L2) | 15.0 | 15.0 | 15.0 | 15.0 | 0.5 |
| Activation Function | ReLU | ReLU | ReLU | ReLU | Tanh |
| Actor/Critic Hidden Dims | [128, 128] | [128, 128] | [128, 128] | [128, 128] | [64 64] |
| $f_\theta$ Hidden Dims | [64, 64] | N/A | N/A | N/A | N/A |
| $g_\theta$ Hidden Dims | [64, 64] | N/A | N/A | N/A | N/A |

For all baselines and ablations, we use PPO (Schulman et al., 2017; Kostrikov, 2018) as the RL training algorithm. Table 4 and Table 5 list the hyperparameters related to architectures and PPO training for Kuhn Poker and Overcooked respectively.

We keep the original hyperparameters for GSCU on Kuhn Poker. For Kuhn Poker, the training budget for all algorithms except GSCU is 5 million steps, while for GSCU the embedding learning takes 1 million episodes and conditional RL takes 1 million episodes. For Overcooked, the training budget for all algorithms except GSCU is 30 million steps, while for GSCU the embedding learning takes 2 million steps and the conditional RL takes 30 million steps. The training of PAIC takes $\sim 12$ hours with $\sim 80$ processes on a single Titan Xp GPU. Both the PAIC actor $\pi_\theta(a|o, \chi_\theta(C))$ and critic take concatenated observation and encoder output as the input.

For algorithms using RNN, including Generalist, LIAM, and LIAMX, the RNN is implemented as a single-layer GRU with 128 hidden units. The RNN is trained using back-propagation through time (BPTT) with gradients detached every 20 steps. Actor and critic share the same RNN, as well as the hidden layers before the RNN.

For methods with auxiliary tasks, there is an additional loss accompanying the main RL loss, computed using the same mini-batch as used in RL training. For PAIC, the auxiliary loss is used with a weight 1.0. For LIAM, the auxiliary loss is used with weight 1.0 for both action and observation prediction. For LILI, the context is the last episode, as specified in the paper. The auxiliary loss is used with weight 1.0 for both reward and next observation prediction.

The coefficient for PAIC's exploration reward decays from $c_{\text{init}}$ to 0 in $M$ steps. $c_{\text{init}} = 0.2$ for Overcooked and 0.01 for Kuhn Poker, while $M = 2.5 * 10^7$ for Overcooked and $4 * 10^6$ for Kuhn Poker. Additionally, we warm up the context encoder for $M_w$ steps using the auxiliary loss only without RL loss. $M_w = 10^6$ for Overcooked and $10^5$ for Kuhn Poker.

# E    MULTI-AGENT MIXED COOPERATIVE-COMPETITIVE ENVIRONMENT

## E.1    ENVIRONMENT DESCRIPTION

Here we introduce an environment with multiple peers, where some peers cooperate with the ego agent and others compete with it. We use a modified version of the predator-prey scenario from the Multi-agent Particle Environment (Lowe et al., 2017) commonly used in the MARL literature. As illustrated in Fig. 8(a), the environment features two predators (red circles, darker one of which is the ego agent), two preys (green circles), and multiple landmarks (grey and blue circles). The predators are tasked with chasing the prey while the prey escape from the predators. Furthermore, the predators are required to collaborate such that all of the prey are covered by predators (see the Reward section

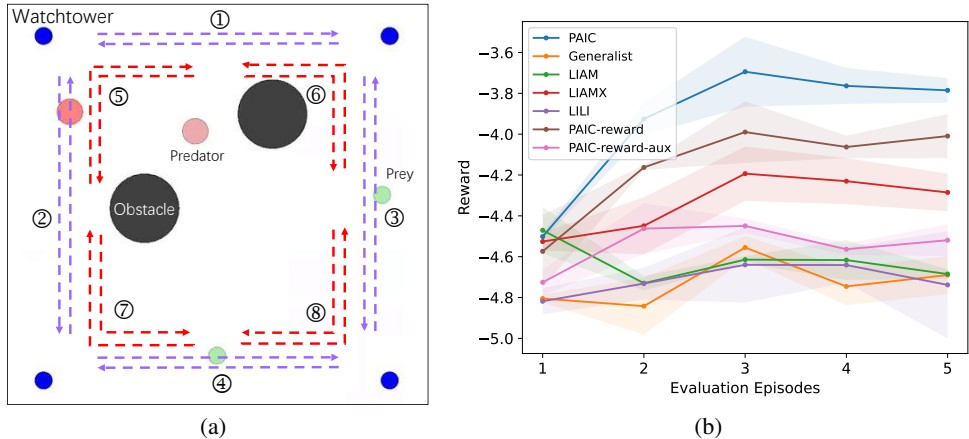

Figure 8: An example of the predator-prey environment (a) and the online adaptation results, including ablations (b).

below). Each episode lasts for at most $40$ steps. If all preys have been touched by predators, the episode terminates immediately.

To make the task harder, we additionally introduce partial observability and four watch towers (blue circles, corners of the figure). The ego agent can only observe agents and landmarks within its observation radius, which is set to $0.2$ throughout the experiments. The ego agent may choose to navigate to the watch tower for better observability. During its contact with any of the watch towers, the ego agent can observe all the agents and landmarks in the environment.

**Observation space.** The observation space is a 37-dimensional vector, consisting of the positions and velocities of the agents and the positions of the landmarks. An additional 0/1 sign is added for every landmark and every agent other than the ego agent to indicate if the entity is currently visible by the ego agent. Invisible entities have the sign, positions, and velocities set to $0$. All positions, excluding that of the ego agent, are relative to the ego agent.

**Action space.** We use the discrete action space of MPE, with 5 actions corresponding to moving left/right/up/down and standing still.

**Reward.** The predators share a common reward that encourages them to collaborate and cover all the preys. Specifically, denote $A$ as the set of all predators and $B$ as the set of all preys, the reward for predators at each time step is given as

$$-c \sum_{b \in B} \min_{a \in A} d(a, b)$$

where $d$ is the Euclidean distance function, $c = 0.1$. Intuitively, this reward allows the predators to divide-and-conquer such that for every prey there is a predator nearby.

### E.2 PEER POOL GENERATION

For the predator peer, we design policies that have a preference towards a specific prey. The predator peer will always chase the preferred prey under full observation.

For the prey peers, we construct $8$ different patterns (Fig. 8(a), dotted lines), where each prey peer moves back and forth along a preferred path. We divide the set of paths into a train set (blue dotted lines, 1-4) and a test set (red dotted lines, 5-8). The final train peer pool is generated by sampling different combinations of 1 predator peer and 2 train prey peers, while the test peer pool samples combinations of 1 predator peer and 2 test prey peers. As a result, during online adaptation, the policies of all prey peers are unseen to the ego agent. We sample 16 combinations for training and 24 combinations for testing.

Table 6: The average reward of PAIC, baselines, and ablations over 5 online test episodes in Predator-Prey. PAIC-r is PAIC without exploration reward, PAIC-r-a is PAIC without exploration reward, and peer identification loss.

| PAIC | Generalist | LIAM | LIAMX | LILI | PAIC-r | PAIC-r-a |
|---|---|---|---|---|---|---|
| $-3.93 \pm 0.04$ | $-4.73 \pm 0.03$ | $-4.62 \pm 0.02$ | $-4.34 \pm 0.08$ | $-4.71 \pm 0.09$ | $-4.16 \pm 0.07$ | $-4.54 \pm 0.03$ |

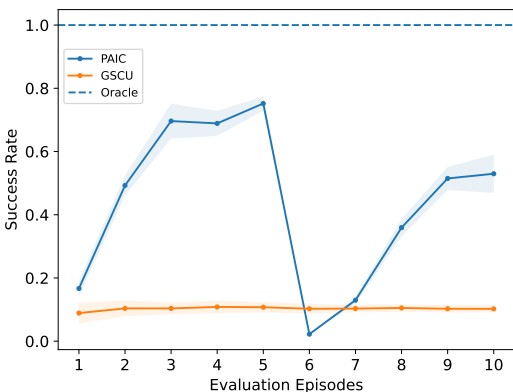

Figure 9: Results of dynamic peer on Overcooked. After the peer change at episode 6, PAIC detects the change and clears the context, allowing performance to recover, while GSCU fails.

### E.3 EXPERIMENT RESULTS AND ABLATIONS

In this environment, we set the context length to $N_{ctx} = 5$ for all the experiments. See Fig. 8(b) and Tab. 6 for the results of PAIC, baselines, and ablations. PAIC outperforms all the baselines and ablations, improving along the interaction episodes. We note that LIAMX equipped with cross-episode contexts and its prediction tasks can outperform both LIAM and Generalist, but not PAIC, demonstrating the superiority of our method. For ablations, removing the exploration reward from PAIC (PAIC-reward) decreases performance, while further removing the peer identification loss (PAIC-reward-aux) sinks it below LIAMX.

### F DYNAMIC PEER EXPERIMENT

We further conducted an experiment with peers that changed during the online interaction episodes. We run the experiment on Overcooked for 10 episodes. The ego agent needs to collaborate with two different peers in the first 5 episodes and the last 5 episodes without knowing when the peer change takes place. While the PAIC agents are trained against static peers that never change, we use drops in the evaluation metric as an indicator of peer changes and clear the context when such changes happen. Formally, denote the evaluation metric for episode $i$ as $R_i$, then we detect a peer change at $i$ iff

$$R_i < c_{th} * \max_{j \leq i} R_j + (1 - c_{th}) * \min_{j \leq i} R_j$$

where $c_{th} \in [0, 1]$ is a threshold coefficient. $c_{th} = 1$ is the most aggressive (clears context whenever performance drops) and $c_{th} = 0$ is the most conservative (never clears the context).

The results are in Fig. 9. PAIC successfully recovers after the change of peer agent at episode 6, while GSCU falls back to the conservative policy, yielding a stable but bad performance.

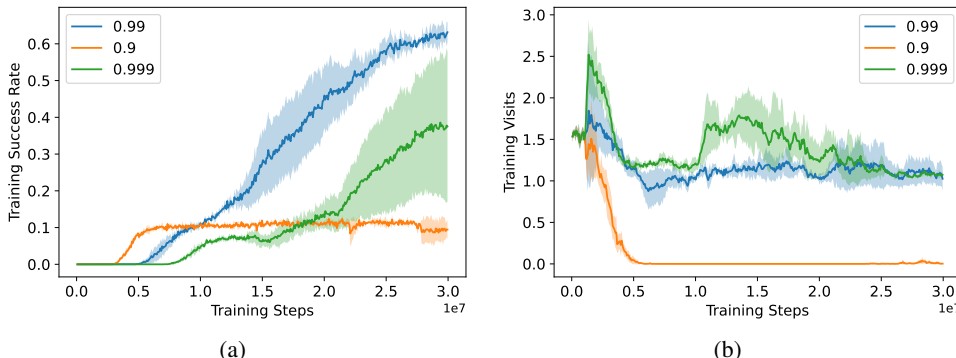

Figure 10: Training success rates (a) and number of visits to the lower room (b) for the discount factor ablations on Overcooked. Our discount factor of choice, $\gamma = 0.99$, performs best. $\gamma = 0.9$ is short-sighted and does not exhibit exploration behaviors, while $\gamma = 0.999$ introduces training instabilities.

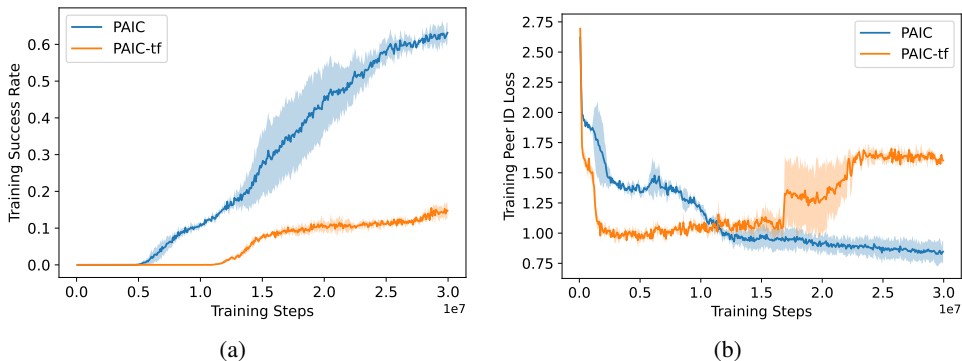

Figure 11: Training success rates (a) and peer ID loss (b) for the Transformer encoder ablations on Overcooked.

## G  ADDITIONAL ABLATIONS

We present additional ablations on the discount factor and Transformer encoder architecture in this section.

### G.1  IMPACT OF DISCOUNT FACTOR

We conduct an ablation in Overcooked for the impact of the discount factor. Fig. 10(a) and 10(b) presents the training success rates and number of visits to the lower room. The number of visits to the lower room reflects the exploratory tendencies of the agent, since this action only reveals information (behaviors of the peer agent in the lower room) to the agent instead of generating imminent environment rewards. As seen in Fig. 10, $\gamma = 0.9$ leads to a short-sighted policy that quickly converges to a local minima, even in the presence of an exploration reward. A larger $\gamma = 0.999$ leads to unstable training, as larger discount factors are known to increase the variance of value estimates.

### G.2  IMPACT OF TRANSFORMER ENCODER

We design a bi-level transformer encoder to compare with the MLP encoder we used in our primary experiments. We construct two transformer encoders for encoding episode-level information and context-level information, respectively. Formally, the encoding for episode $n$ is

$$z_n^1 := Enc^e((o_{n,1}^1, a_{n,1}^1), (o_{n,2}^1, a_{n,2}^1), \ldots, (o_{n,T_n}^1, a_{n,T_n}^1))$$

while the encoding for the whole context $C$ is

$$z^1 := Enc^c(z_1^1, z_2^1, \ldots, z_N^1)$$

where $Enc^e, Enc^c$ are transformer encoders for the episode- and context-level, followed by average pooling over the sequence dimension. We apply learnable positional embedding and standard regularization techniques like Dropout to the encoder. We use hidden dimension and feed-forward dimension of 128, Dropout coefficient of 0.5, 4 heads for the multi-head attention, and a single layer of transformer encoder in each of $Enc^e$ and $Enc^c$.

As shown in Fig. 11, PAIC-tf is greatly outperformed by the standard PAIC MLP encoder. The reason for this performance, we hypothesize, is because the transformer encoder may overfit to the peer identification task. In Fig. 11(b), it can be seen that the peer ID loss of PAIC-tf drops faster than PAIC initially, but slowly increases after that, until an abrupt change in the middle of the training. In contrast, the loss for PAIC drops consistently along the training procedure. It is worth noting that our peer identification task and exploration reward is not limited to a specific kind of encoder architecture, and that we choose MLP with average pooling based on empirical performance.

