# OpenReview forum: "Learning to Explore with In-Context Policy for Fast Peer Adaptation"
_ICLR.cc/2024/Conference — Submitted to ICLR 2024_

### Official Review · Reviewer_gKAo · 2023-10-30

**Soundness:** 3 good
**Presentation:** 3 good
**Contribution:** 2 fair
**Rating:** 5
**Confidence:** 4

**Summary:**

In MARL settings where one agent interacts with one peer, this paper proposes learning a context-conditioned policy for the agent to adapt to different peers without knowing their type. The focus here is on being able to balance exploration-exploitation in the agent's interaction with the peer. To achieve this, the agent learns to predict the peer's type as an auxiliary task and jointly trains the type prediction and policy networks so that the agent's policy prefers exploratory actions when there is uncertainty about the peer's type; otherwise exploitative actions are preferred to maximize the agent's reward. The authors consider two environments: Overcooked in which the agent-peer interaction is cooperative and Kuhn Poker in which the agent-peer interaction is competitive. In both cases, experimental results indicate that the proposed framework enables the agent to adapt to its peer under partial observability, as well as over short and long horizons.

**Strengths:**

1. The authors have presented a thorough discussion of the related work in their problem setting and clearly  highlighted connections to and differences from prior work. A number of baselines from prior work have been considered in Sec 4.1 to demonstrate the superior performance of the proposed approach.

2. This paper's approach of jointly training auxiliary peer type prediction network with the context conditioned RL policy and using intrinsic reward to guide the explore-exploit trade-off during training, is simple yet effective in the proposed settings. Although this framework relies on several restrictive assumptions, the authors have also highlighted many of those in Section 5.

**Weaknesses:**

1. The proposed framework depends on the availability of the finite set $\Psi$ of peer types, therefore either a diverse set of peers should be available at training time or the agent might fail to generalize to previously unseen types of peers during evaluation. This is a major limitation of this approach.

2. This work assumes that the policy followed by the peer is stationary and fixed, which makes it a much simpler setting than for example prior work in [1]. Although the authors claim that the one agent - one peer framework followed in the paper will generalize to multiple peers, it perhaps would not be as easy to extend to more complicated settings for example, when multiple peers cooperate with each other to compete against the agent. I suspect this would be the case because the agent only predicts the type of a peer from the set $\Psi$, and it is not practical to assume that all possible interactions between multiple peers can be enumerated in $\Psi$ during training.

[1] Banerjee, A., Phade, S., Ermon, S. and Zheng, S., 2023. MERMAIDE: Learning to Align Learners using Model-Based Meta-Learning. arXiv preprint arXiv:2304.04668.

**Questions:**

1. It would help to add an algorithm box for the evaluation phase - particularly, I am confused whether or not the policy or peer type prediction network is fine-tuned during the adaptation to test agents. What are the differences between training and test settings? This would also help me better understand the results in Fig 4.

2. In Fig 4, is it possible to design a baseline that upper bounds the performance of the learning based approaches? For example, an expert policy that has access to an oracle for the peer type.

3. In Sec 4.4, please clearly specify the definitions of each baseline - Fig 5 labels "PAIC-reward" and "PAIC-reward-aux" but they are not mentioned in the text.

4. In Sec 4.5, last sentence: "While there are minor differences,..." - where is this result shown?

5. In Table 3, for Train 1, why is there a drop in performance for larger pool size N (i.e. N=18 to N=36)?

6. Fig 6 - Is the t-SNE plot for Overcooked? Please make it explicit.

---

> ### Author Response · Authors · 2023-11-22
> **Reply to Reviewer gKAo**
>
> > **Q1: The proposed framework depends on the availability of the finite set Ψ of peer types, therefore either a diverse set of peers should be available at training time or the agent might fail to generalize to previously unseen types of peers during evaluation. This is a major limitation of this approach.**
>
> A1: Generating a diverse pool of agents for zero-shot multi-agent coordination is a widely studied topic[1][2]. It is true that our method can also benefit from a diverse set of peers at training. However, it should be emphasized that PAIC focuses on how to learn a policy to balance exploration and exploitation for adaptation given a training peer pool, which is orthogonal to the diverse pool generation problem. The training pools in our experiments are shared across all methods, and PAIC still outperforms all the baselines and ablations, showing the effectiveness of PAIC. Note that the previous works (GSCU, TrajDi, …) also train the models in a finite set of model pools. Moreover, the types of the test peer policy in the newly added watchtower predator-and-prey are in fact **unseen**. Please refer to Appendix E for more details.
>
> > **Q2: More details on evaluation. Whether or not the policy or peer-type prediction network is fine-tuned during the adaptation to test agents. What are the differences between training and test settings?**
>
> A2: During evaluation, all networks are frozen without any fine-tuning. In fact, peer identification is only an auxiliary task for boosting the training of the context encoder. During testing/evaluation, the peer identification networks are not used at all. We only use the context encoder to obtain the latent representation of the peer. There are 2 differences between training and test procedures.  One is that the testing and training peer pool (policies) are different. The other difference is that all networks are fixed and the peer identification head is dropped during training.
>
> > **Q3: A baseline that upper bounds the performance of the learning-based approaches?**
>
> A3: Thanks for your suggestion! We have added a referential upper-bound baseline “Oracle” in Figure 4. This baseline has access to the Oracle peer type so that it can be regarded as the “best response” to the peer, making it the upper bound.
>
> > **Q4: Clearly specify the definitions of "PAIC-reward" and "PAIC-reward-aux” in Fig 5.**
>
> A4: Thanks for pointing out this oversight! PAIC-reward means that we reserve the auxiliary task to train the context encoder but remove the auxiliary exploration reward. PAIC-reward-aux means we both remove the auxiliary task and the exploration reward. Figure 5 shows that the auxiliary task can slightly boost the context encoder and the exploration reward significantly contributes to the performance. We have added more explanations in Sec 4.4 to clarify the terms.
>
> > **Q5: In Sec 4.5, the last sentence: "While there are minor differences,..." - where is this result shown?**
>
> A5: We apologize for the misleading statement. In fact, what we wanted to claim is that PAIC agents trained on different training pools surpass the baseline performance in Table 2, where all methods are trained on the base common training pool (18 policies). We have modified the statement to make it clearer. Thanks for your suggestion!
>
> > **Q6: In Table 3, for Train 1, why is there a drop in performance for larger pool size N (i.e. N=18 to N=36)?**
>
> A6: Good point! This is caused by the random seed. We rerun the experiments on 3 seeds and report the average results in the Table. 3, showing the drop disappeared in N=36. We have updated the results in the revision.
>
> > **Q7: Fig 6 - Is the t-SNE plot for Overcooked? Please make it explicit.**
>
> A7: Thanks for pointing out this oversight! This t-SNE plot is for Overcooked. We have modified the caption of Fig 6 to clarify it.
>
> Reference:
>
> [1] Andrei Lupu, Brandon Cui, Hengyuan Hu, and Jakob Foerster. Trajectory diversity for zero-shot coordination. In International conference on Machine Learning, pp. 7204–7213. PMLR, 2021.
>
> [2] Rujikorn Charakorn, Poramate Manoonpong, and Nat Dilokthanakul. Generating diverse cooperative agents by learning incompatible policies. In International Conference on Learning Representations, 2023.

---

### Official Review · Reviewer_cQ6z · 2023-10-30

**Soundness:** 2 fair
**Presentation:** 4 excellent
**Contribution:** 2 fair
**Rating:** 5
**Confidence:** 4

**Summary:**

This paper presents an approach, called Fast Peer Adaptation with In-Context policy (PAIC), for multi-agent settings where agents need to quickly adapt to diverse peer behaviors in both cooperative and competitive scenarios. The main contributions of the paper are as follows:

1. The paper recognizes a different method to identify peer patterns, and adapt accordingly based on the history.

2. PAIC tries to balance exploration and exploitation, allowing agents to optimize their performance during peer adaptation. It promotes exploratory actions when the context is uncertain. An intrinsic reward mechanism is introduced based on peer identification accuracy.

3. The proposed method is evaluated in both competitive (Kuhn Poker) and cooperative (Overcooked) environments, demonstrating faster adaptation and improved performance compared to existing methods when facing novel peers.

**Strengths:**

1. The paper is well-prepared, with clear writing and figures.

2. The paper introduces an important question: exploration in ad-hoc cooperation.

3. The reviewer was impressed by the strong empirical performance of the proposed method.

**Weaknesses:**

1. (Major, about "in-context learning")

1.1 The reviewer failed to discern the rationale behind the authors' proposal of the in-context policy as a novel concept. In-context policies fundamentally rely on previous trajectories, a characteristic shared by nearly all multi-agent policies. What distinguishes this approach is the utilization of trajectories from various instances. However, if an RNN or a Transformer is used to represent agents' policies, it might also remember information from previous episodes.


1.2 The omission of peer information in these trajectories could potentially misguide the trajectory embedding.

1.3 The justification for employing a Multi-Layer Perceptron (MLP) to encode trajectories merits a more thorough validation. As the authors suggest, utilizing MLP treats the context as a collection of state-action pairs, which disregards the intra- and inter-episode temporal order. This oversight could certainly have a negative impact on context encoding performance. Furthermore, the reviewer does not concur that alternative network architectures are incapable of capturing long-term dependencies and mitigating overfitting. Transformers, for instance, have been widely adopted in the encoding of multi-agent trajectories.

2. Why identifying peers can be regarded as an intrinsic reward that encourages exploration?

**Questions:**

1. The the context only include ego information? How can a peer be identified without its information?

2. What if a transformer is used for context encoding? Would it be better than an MLP?

3. In Figure 5, what is the difference between PAIC-reward and PAIC-reward-aux?

4. More ablations on other scenarios are expected.

---

> ### Author Response · Authors · 2023-11-22
> **Reply to Reviewer cQ6z**
>
> > **Q1: The reviewer failed to discern the rationale behind the authors' proposal of the in-context policy as a novel concept. In-context policies fundamentally rely on previous trajectories, a characteristic shared by nearly all multi-agent policies. What distinguishes this approach is the utilization of trajectories from various instances. However, if an RNN or a Transformer is used to represent agents' policies, it might also remember information from previous episodes.**
>
> A1: We agree that in-context policy is a kind of method that utilizes history trajectories, but the concept of in-context policy is not the core contribution of our work. As you mentioned, both PAIC and the existing RNN/Transformer-based methods can encode context information for downstream tasks. However, PAIC has a significant advantage over the previous methods in terms of learning to explore the strategy of unknown peers.  PAIC introduces a peer identification auxiliary task that serves two purposes: first, it facilitates the training of the context encoder by providing a supervised signal; second, it generates intrinsic rewards that mitigate the sparse reward problem and encourage exploration. The intrinsic rewards are based on the information gained from the states, which means PAIC can actively seek informative states to model the peers better. Therefore, PAIC can adapt to unknown peers faster and more accurately.
>
> >  **Q2: The context only include ego information? How can a peer be identified without its information?**
>
> A2: Yes, the context does only include ego information since we consider the partially observable setting. For example in the kuhn poker game, the hand is not revealed until the showdown stage. The peer information is partially revealed in the ego observation, so the agent should actively explore to gain enough information to identify the strategy of the peer.  This is also the reason why we introduce the auxiliary reward to encourage active exploration behavior, making PAIC different from other recurrent models.
>
> > **Q3: What if a transformer is used for context encoding? Would it be better than an MLP?**
>
> A3: We tried to replace MLP with Transformer for the context encoder. To visualize the comparison between MLP and Transformer-based context encoder, we add more curves and data in Appendix G. It is shown that for the Transformer-based context encoding, the peer identification loss quickly drops before a rise, while the loss of MLP version steadily decreases. This is because the Transformer-based context encoder quickly overfits the context distribution in the early stage of training. As the learning proceeds, the agent policy changes and leads to a different observation (context) distribution, causing the identification loss to rise. As for the MLP-based encoder, it does not overfit the context in the early stage and progressively improves throughout training. The success rate curve in Figure 11(a) also supports our claim. Please refer to Appendix G.2 for more details.
>
> > **Q4: In Figure 5, what is the difference between PAIC-reward and PAIC-reward-aux?**
>
> A4: Thanks for pointing out this oversight! PAIC-reward means that we reserve the auxiliary task to train the context encoder but remove the auxiliary exploration reward. PAIC-reward-aux means we both remove the auxiliary task and the exploration reward. Figure 5 shows that the auxiliary task can slightly boost the context encoder and the exploration reward significantly contributes to the performance. We have added more explanations in Sec 4.4 to clarify the terms.
>
> > **Q5: Why identifying peers can be regarded as an intrinsic reward that encourages exploration?**
>
> A5: As we mentioned in Answer 1 for Question 1, the ego agent can only partially observe the peer. Identifying the peer is difficult without enough information provided by active exploration. Therefore, exploration is the prerequisite for successful identification. In this way, peer identification can serve as an auxiliary task and provide an intrinsic reward that encourages exploration.
>
> > **Q6: More ablations on other scenarios are expected.**
>
> A6: We further conduct an ablation study in the new watchtower predator-and-prey environments, showing consistent results as the previous environments. Please refer to the Appendix. E for more details. The results in the following Table demonstrate the effectiveness of the PAIC in the new environment, compared with baseline methods.
> |   PAIC   | Generalist | LIAM | LIAMX | LILI | PAIC-reward | PAIC-reward-aux |
> | :--: | :--: | :--: | :--: | :--: | :--: | :--: |
> | **-3.934 $\pm$ 0.038** | -4.728 $\pm$ 0.031 | -4.623 $\pm$ 0.018 | -4.337 $\pm$ 0.079 | -4.714 $\pm$ 0.094 | -4.160 $\pm$ 0.065 | -4.544 $\pm$ 0.033 |

---

### Official Review · Reviewer_VfPt · 2023-11-01

**Soundness:** 3 good
**Presentation:** 3 good
**Contribution:** 3 good
**Rating:** 6
**Confidence:** 3

**Summary:**

This paper proposes a novel end-to-end method called Fast Peer Adaptation with In-Context Policy (PAIC) for training agents to adapt to unknown peer agents efficiently. PAIC learns an in-context policy that actively explores the peer's policy, recognizes its pattern, and adapts to it. The agent is trained on a diverse set of peer policies to learn how to balance exploration and exploitation based on the observed context, which is the history of interactions with the peer. The paper introduces an intrinsic reward based on the accuracy of the peer identification to encourage exploration behavior. The method is evaluated on two tasks involving competitive (Kuhn Poker) or cooperative (Overcooked) interactions with peer agents, demonstrating faster adaptation and better outcomes than existing methods.

**Strengths:**

* PAIC achieves faster adaptation and better outcomes compared to existing methods in both competitive and cooperative environments.
* The introduction of an intrinsic reward based on the accuracy of peer identification encourages exploration behavior, which is crucial for efficient adaptation.
* The method is evaluated on two diverse environments, Kuhn Poker and Overcooked, showcasing its effectiveness in both competitive and cooperative settings.

**Weaknesses:**

* The paper only considers purely cooperative and competitive environments. It would be interesting to see whether PAIC can handle more complex mixed-motive environments.
* The paper assumes that the peer agent does not update its policy during test time. However, in the real world, peers may be able to tune their policies online, which could pose a challenge for PAIC.

**Questions:**

* How does PAIC perform in more complex mixed-motive environments, where agents have to balance their own interests and the collective welfare?
* Can PAIC adapt to non-stationary peers, where the peer agent is also updating its policy during test time?
* What are the implications of PAIC for human-AI interaction, and how can it be evaluated in real-world settings involving human peers?
* Are there any potential ethical considerations or challenges when applying PAIC to real-world scenarios, such as human factors and user feedback?

---

> ### Author Response · Authors · 2023-11-22
> **Reply to Reviewer VfPt**
>
> > **Q1: How does PAIC perform in more complex mixed-motive environments, where agents have to balance their own interests and the collective welfare?**
>
> A1: PAIC can perform well in the mixed-motivate multi-agent environment. Here, we introduce a **watchtower predator-and-prey environment** for further evaluation. In this environment, the prey has different moving patterns across training and testing. The predator only can observe the states in a small range of circles. There are four watchtowers in the corner. When the predator is in the watchtower, it can get the global state. In that case, the predator has to balance chasing the closest prey (own interests) and going to the towner to make the other prey visible to the team (collective welfare). More details about the environment are available in the Appendix. E. The results in the following Table demonstrate the effectiveness of the PAIC in such environments, compared with baseline methods.
>
> > **Q2: Can PAIC adapt to non-stationary peers, where the peer agent is also updating its policy during test time?**
>
> A2: We conduct an additional experiment to evaluate the performance of PAIC in adapting to non-stationary peers in Overcooked. The test context length is 10 episodes, where the ego agent is paired with one peer in the first 5 episodes and another peer in the next 5 episodes. The PAIC agent is not informed of this peer switching test procedure, so we can test how fast the PAIC agent can detect the peer change and adapt to the new peer. The results in Appendix F show that although there is a drop in success rate in the first episode of the new peer, the PAIC agent manages to rapidly adapt to the new peer within a few interactions. Please refer to Appendix F for more details.
>
> > **Q3: What are the implications of PAIC for human-AI interaction, and how can it be evaluated in real-world settings involving human peers?**
>
> A3: Good question! In fact, one of the potentials of PAIC is to enhance human-AI interaction. In the future, when AI agents are spread across the world, they may encounter unfamiliar human users every day. The AI agents need to quickly identify the behavior patterns and the internal values of human users so as to serve them, where PAIC can help. As for evaluating PAIC in real-world settings involving human peers, we can consider the scenario where human players need an AI agent partner to cooperate or compete with. We can first collect some human trajectories to train human proxy agents for the training peer pool. Then we can pair the trained PAIC agent with human players in the game, for example Overcooked[1], to see how the PAIC agent adapts to human players.
>
> Reference: [1] Strouse, D.J., McKee, K., Botvinick, M., Hughes, E. and Everett, R., 2021. Collaborating with humans without human data. *Advances in Neural Information Processing Systems*, *34*, pp.14502-14515.
>
> > **Q4: Are there any potential ethical considerations or challenges when applying PAIC to real-world scenarios, such as human factors and user feedback?**
>
> A4: One potential consideration and challenge may be how to construct a representative peer pool for training. For real-world scenarios involving human peers, we need to collect sufficient human behaviors to generate a representative training peer pool. However, such a process may involve collecting sensitive personal information and may pose ethical issues. For example, we may need to obtain the consent of the human users to record their trajectories and use them for training the AI agents. We may also need to ensure the privacy and security of the collected data and prevent any unauthorized access or misuse. Moreover, we may need to consider the diversity and fairness of the training peer pool and avoid any bias or discrimination that may affect the performance and behavior of the AI agents. These are some of the ethical considerations and challenges that we need to address when applying PAIC to real-world scenarios.

---

### Official Review · Reviewer_wMXp · 2023-11-06

**Soundness:** 2 fair
**Presentation:** 1 poor
**Contribution:** 2 fair
**Rating:** 3
**Confidence:** 4

**Summary:**

The authors introduce a context encoder, which is enhanced with an additional task to distill concise information from sequences of {observation, action}. Subsequently, policies are influenced by this hidden context data to promote more desirable actions. The auxiliary task aids the context encoder in categorizing the observed sequences into a finite set of types. By minimizing the loss associated with the auxiliary task, the context encoder effectively learns to adapt to various types of agents, without the need to predefine agent types as seen in prior studies. While the paper lacks a detailed discussion on the convergence properties of PAIC, empirical evaluations in Kuhn Poker and Overcooked environments reveal superior performance compared to some relevant baseline methods.

**Strengths:**

Originality
The paper represents a compelling fusion of context encoders with an auxiliary task that resembles classification. The paper effectively highlights the distinctions when compared to a closely related study.

Quality
The paper incorporates motivating examples, and it includes an ablation study where various PAIC-specific parameters are examined.

Clarity
The encoder and the auxiliary task (peer identification) and the extrinsic reward defined therein, are explained well, and limitations have been identified.

Significance
The achieved performance significantly outperforms the baseline methods, and the in-context policy learning framework seems well-suited for scenarios involving two agents, although the paper does not delve into the convergence behavior of PAIC in such extended settings.

**Weaknesses:**

The paper faces several challenges in its related work and critical analysis. While the idea of utilizing latent representations of trajectories is not novel, and concepts like cross-entropy loss and auxiliary tasks are commonly used in various classification problems, including class-incremental continual learning, the paper falls short in discussing how these methods can be effectively extended to multi-agent settings. This lack of extension and exploration makes it challenging to gauge the originality and innovation of PAIC in comparison to existing approaches.

Moreover, the paper does not address fundamental questions and considerations that are crucial for understanding its applicability and limitations. It fails to provide insights into how PAIC can be extended to scenarios with more than two agents, raising questions about adaptation in situations where agents significantly diverge from each other. Discount factors and their impact on PAIC are not discussed, which weakens the quality of the evaluation section.

Experiment details can be improved. Important aspects, such as how the baselines were fine-tuned, the training process of the Generalist against all peers are left unexplained. The impact of varying N_test values and the absence of a discussion on convergence properties further hinder a comprehensive understanding of PAIC. The paper does not provide theoretical evidence supporting its claim of solving the partially observable stochastic games (POSG), and the problem formulation lacks in-depth derivations and convergence analyses. Without addressing these issues and discussing how PAIC can be extended to more complex scenarios or under what conditions it converges, it is challenging to assess the significance and practicality of PAIC.

**Questions:**

How does PAIC scale beyond two agents?
How do some other recurrent models compare against PAIC? What is the expected/observed advantages/disadvantages of the recurrence?

---

> ### Author Response · Authors · 2023-11-22
> **Reply to Reviewer wMXp**
>
> > **Q1: What makes PAIC different from existing methods that utilize recurrent models to encode context information?**
>
> A1: Both PAIC and the existing methods can use recurrent models or other sequential encoders to encode context information for downstream tasks. However, PAIC has a significant advantage over the previous methods in terms of learning to explore the strategy of unknown peers.  PAIC introduces a peer identification auxiliary task that serves two purposes: first, it facilitates the training of the context encoder by providing a supervised signal; second, it generates intrinsic rewards that mitigate the sparse reward problem and encourage exploration. The intrinsic rewards are based on the information gained from the states, which means PAIC can actively seek informative states to model the peers better. Therefore, PAIC can adapt to unknown peers faster and more accurately.
>
> > **Q2: How can PAIC be extended to multi-agent settings?**
>
> A2: To evaluate the generalization of PAIC to multi-agent settings, we introduce a novel **watchtower predator-and-prey environment** as a challenging benchmark.  In this environment, there are two predators and two prey with different moving patterns in the training and testing phases. The predators have limited observation range and can only see the states within a small circle. However, there are four watchtowers at the corners of the environment that can provide global observation. When a predator reaches a watchtower, it can obtain the global state. In that case, the predator has to balance chasing the closest prey (own interests) and going to the towner to make the other prey visible to the team (collective welfare). More details about the environment are available in the Appendix. E. The results in the following Table show that PAIC outperforms the baseline methods in this environment, demonstrating its effectiveness in adapting to multi-agent settings.
>
> |   PAIC   | Generalist | LIAM | LIAMX | LILI | PAIC-reward | PAIC-reward-aux |
> | :--: | :--: | :--: | :--: | :--: | :--: | :--: |
> | **-3.934 $\pm$ 0.038** | -4.728 $\pm$ 0.031 | -4.623 $\pm$ 0.018 | -4.337 $\pm$ 0.079 | -4.714 $\pm$ 0.094 | -4.160 $\pm$ 0.065 | -4.544 $\pm$ 0.033 |
>
>
> > **Q3: What is the impact of discount factors on PAIC?**
>
> A3: We conduct additional ablation experiments to study to impact of discount factors. The learning curve in Figure 10(a) shows that the discount factor gamma is indeed crucial for PAIC. When gamma is too low (0.9), it diminishes the long-term success reward and leads to the disappearance of exploration behavior, significantly reducing performance. When gamma is too high (0.999), it causes unstable training and also a drop in performance. Therefore, the discount factor gamma is an important hyper-parameter in PAIC learning. Please refer to Appendix G.1 for more details.
>
> > **Q4: Some training details are left to clarify. How are the baselines fine-tuned? What is the training process of the Generalist against all peers?**
>
> A4: PAIC employs the general training pipeline, previously used in LILI, LIAM, LIAMX, and Generalist, in Algorithm 1. In our experiments, all methods use the same training peer pool as PAIC. The main difference is that these methods have different auxiliary tasks and do not use the exploration reward that PAIC employs. We use the term ego agent to refer to the agent being trained (PAIC, baseline, or Generalist).  During each training epoch, each peer policy from the pool is paired with the ego agent for a fixed length of context (5 episodes for example). The ego agent is trained by the corresponding algorithm to maximize its return in the given context length.  For Generalist, it is an RNN policy that maintains its hidden state across episodes, allowing it to model cross-episode context. The Generalist policy can be viewed as an RNN variant of PAIC-reward-aux. The training procedure of GSCU is quite different from other methods, which can be found in the original paper. We provide more training details in Appendix D.
>
> > **Q5: What is the impact of varying N_test values?**
>
> A5: Sorry, there is no variable named N_test in this paper. N denotes the number of episodic trajectories in the context, and N_ctx indicates the maximum number of episodes to adapt, which is equal to the maximum length of the context. We guess you are referring to the maximum number of episodes we set in the testing phase, which is the same as the training settings in our experiments. The curves in Fig. 4 also show the performance in the adaption process with the increasing of the test-time episodes.
>
> > **Q6: Lack of theoretical evidence supporting its claim of solving the partially observable stochastic games (POSG)**
>
> A6: Sorry, we did not mention the partially observable stochastic games in our manuscript. Please check if this comment is relevant to our paper.

---

### Meta-Review · Area_Chair_FCnw · 2023-12-09

**Metareview:**

This paper focuses on MARL problems by proposing a context-conditioned policy to adapt to different agents / peers. Experiment in Overcooked and Kuhn Poker show improved results. The reviewers appreciate the simplicity of the approach, diverseness of environments, positive results and clarity of writing. However, several concerns have also been raised from the novelty of in-context policies, restrictive assumptions on the peers, and stationary of the peers. On closer look at the reviews, we found that one of the reviews missed out on key details and we omit that review in evaluating this paper. However, the remaining reviewers are unconvinced on the potential impacts of this work given the restrictive assumptions. I would hence recommend resubmitting this work with the additional (third) task and discussion of assumptions brought up front.

**Justification For Why Not Higher Score:**

Limited novelty, and restrictive assumptions on the peer / agents used for the MARL setting.

**Justification For Why Not Lower Score:**

N/A

---

### Decision · Program_Chairs · 2024-01-16

Reject